# Natural Green Spaces, Sensitization to Allergens, and the Role of Gut Microbiota during Infancy

Vienna Buchholz,[a] Sarah L. Bridgman,[a] Charlene C. Nielsen,[a] Mireia Gascon,[b,c,k] Hein M. Tun,[d,e] Elinor Simons,[f] Stuart E. Turvey,[g] Padmaja Subbarao,[h] Tim K. Takaro,[i] Jeffrey R. Brook,[j] James A. Scott,[j] Piush J. Mandhane,[a] Anita L. Kozyrskyj[a]

[a]Department of Pediatrics, University of Alberta, Edmonton, Alberta, Canada

[b]Barcelona Institute for Global Health (ISGlobal), Barcelona Biomedical Research Park (PRBB), Barcelona, Spain

[c]CIBER Epidemiología y Salud Pública (CIBERESP), Madrid, Spain

[d]The Jockey Club School of Public Health and Primary Care, The Chinese University of Hong Kong, Hong Kong SAR, People's Republic of China

[e]Li Ka Shing Institute of Health Sciences, Faculty of Medicine, The Chinese University of Hong Kong, Hong Kong SAR, People's Republic of China

[f]Section of Allergy & Immunology, Department of Pediatrics & Child Health, University of Manitoba and Children's Hospital Research Institute of Manitoba, Winnipeg, Manitoba, Canada

[g]Department of Pediatrics, BC Children's Hospital, The University of British Columbia, Vancouver, British Columbia, Canada

[h]Department of Pediatrics and Physiology, Hospital for Sick Children, University of Toronto, Toronto, Ontario, Canada

[i]Faculty of Health Sciences, Simon Fraser University, Burnaby, British Columbia, Canada

[j]Dalla Lana School of Public Health, University of Toronto, Toronto, Ontario, Canada

[k]Universitat Pompeu Fabra (UPF), Barcelona, Spain

**ABSTRACT** The environment plays an instrumental role in the developmental origins of health and disease. Protective features of the environment in the development of asthma and atopy have been insufficiently studied. We used data from the CHILD (Canadian Healthy Infant Longitudinal Development) Cohort Study to examine relationships between living near natural green spaces in early infancy in Edmonton, AB, Canada and the development of atopic sensitization at 1 year and 3 years of age in a cohort of 699 infants, and whether these associations were mediated by infant gut microbiota (measured using 16s V4 amplicon sequencing) at 4 months. The Urban Planning Land Vegetation Index (uPLVI) map of the City of Edmonton was used to assess infants' exposure to natural spaces based on their home postal codes, and atopic sensitization was assessed using skin prink testing (SPTs) for common food and inhalant allergens. Our findings suggest there is a protective effect of natural green space proximity on the development of multiple inhalant atopic sensitizations at 3 years (odds ratio = 0.28 [95% CI 0.09, 0.90]). This relationship was mediated by changes to Actinobacteria diversity in infant fecal samples taken at 4 months. We also found a positive association between nature proximity and sensitization to at least one food or inhaled allergen; this association was not mediated by gut microbiota. Together, these findings underscore the importance of promoting natural urban green-space preservation to improve child health by reducing atopic disease susceptibility.

**IMPORTANCE** Our findings highlight the importance of preserving natural green space in urban settings to prevent sensitization to environmental allergens and promote early-life gut microbiota pathways to this health benefit. These findings support a mediating role of gut microbiome compositions in health and disease susceptibility. This study used unique, accurate, and comprehensive methodology to classify natural space exposure via a high-resolution topographical map of foliage subtypes within the City of Edmonton limits. These methods are improvements from other methods previously used to classify natural space exposure, such as the normalized density vegetation index from satellite imagery, which is not able to distinguish anthropogenic from green space. The use of these methods and the associations found between natural green space exposure and atopic sensitization outcomes support their use in future studies. Our findings also provide many avenues for future research including longer term follow up of this cohort

Address correspondence to Anita L. Kozyrskyj, kozyrsky@ualberta.ca.

The authors declare no conflict of interest.

and investigation of a causal role of reduced Actinobacteria diversity on atopic sensitization development.

**KEYWORDS** atopic sensitization, natural space, microbiome, infant, gut microbiota, infants, natural green space, allergy, gut microbiome, inhalant

In this era of climate change, urbanization, land disturbance, and high burdens of noncommunicable disease of the heart and lung, evidence on how the environment affects the origins of health and disease is vital knowledge in structuring public and personal health programs (1). Chronic respiratory diseases, like asthma, still account for an unacceptably high number of deaths in industrialized countries (2). The reduced lung function of respiratory disease often presents in childhood as the atopic phenotype of asthma following an inflammatory immune response to repeated environmental triggers (3–5). Further, atopic conditions such as asthma, atopic dermatitis, and allergic rhinitis are common in Canadian adolescents; they reduce quality of life of those affected, and present substantial health, social and economic costs (6–10). Understanding the developmental origins of these diseases in early life is key to preventing their associated morbidity and mortality.

Bringing natural environments and their microbial ecosystems into our everyday lives is being touted as a real-world intervention to enrich the human microbiome, balance the immune system, and guard against allergy and inflammatory disorders (11). This recommendation builds on landmark findings from the Haahtela research group on nature exposure in Finland, skin microbiota, and atopic disease in older children (11, 12). It is strongly supported by longstanding evidence on the benefits of growing up on a farm in reducing risk of developing atopy (13). It is further supported by more recent findings on the mediating role of the early-life gut microbiome in farm exposure protection against asthma development and on the reversal of airway inflammation following colonization of mice with gut microbes depleted in children with asthma (14–16). While these findings are intriguing, the mechanisms behind them are not fully understood, and most of the world's population does not live in agricultural or rural settings. In this regard, it is important to understand which environmental factors are associated with the risk of developing asthma and other atopic diseases in urban settings.

Systematic reviews of studies on the health benefits of green space exposure in childhood certainly point to the potential of green space in reducing atopic respiratory disease risk (17, 18). However, study results have been inconsistent by geography, green space measures, and study populations (17–21), and some are confounded by air pollution (22). Since vegetation species and local microbial populations vary by geographic area, it becomes important to contextualize the type of green space exposure and document impacts on atopy development in localized areas. In recent work by Nielsen et al. (2020), proximity to natural green spaces in a Canadian urban setting was associated with gut microbial diversity at 4 months of age in infants of the CHILD Cohort Study (23). However, the study did not investigate whether this type of green space, namely, natural green space, was related to future atopic disease and if the gut microbiome mediated atopy outcomes (23). To investigate nature-gut microbiota-atopy pathways, we conducted further analyses using Nielsen et al.'s natural green space mapping index, created from data linkage of infants at the Edmonton site of the CHILD Cohort Study with a one-of-a-kind natural areas map developed by the city of Edmonton. Our objectives were to (i) examine relationships between exposure to natural green spaces in infancy and risk of atopic disease development at ages 1 and 3 years, and (ii) determine if these relationships are mediated by gut microbiota composition at 4 months of age.

## RESULTS

There were 699 infants with assigned natural green space in the city of Edmonton. Of these infants, 530 had complete data on atopic sensitization at age 1 year and 460 had complete data on atopic sensitization at 3 years, according to completed skin prick tests (SPTs) and questionnaires (Table 1 and Table S1). Over half of infants (54.7%) had

**TABLE 1** Summary statistics of the total CHILD Study Edmonton site cohort (N = 699), by natural space <500m from residence, atopic sensitization status at 3 years, and inhalant atopic sensitization status at 3 years[a]

| Participant characteristics | Prevalence Overall N (%) | Natural space <500m from residence | | | ≥1 atopic sensitization at 3 yrs | | | ≥2 inhalant atopic sensitizations at 3 yrs | | |
|---|---|---|---|---|---|---|---|---|---|---|
| | | Yes N (%) | No N (%) | P | Yes N (%) | No N (%) | P | Yes N (%) | No N (%) | P |
| Total | 699 (100) | 366 (54.7) | 303 (45.3) | | 75 (16.3) | 385 (83.7) | | 15 (3.3) | 443 (96.7) | |
| Infant sex | | | | | | | | | | |
| Male | 307 (50.3) | 173 (51.5) | 134 (48.7) | 0.50 | 46 (61.3) | 186 (48.3) | **0.04** | 11 (73.3) | 220 (49.7) | 0.07 |
| Female | 304 (49.8) | 163 (48.5) | 141 (51.3) | | 29 (38.7) | 199 (51.7) | | 4 (26.7) | 223 (50.3) | |
| Missing | 58 | | | | | | | | | |
| Birth wt (grams) | | | | | | | | | | |
| <3,000 | 119 (19.6) | 70 (21.1) | 49 (18.0) | 0.80 | 22 (29.3) | 65 (17.0) | 0.09 | 4 (26.7) | 83 (18.9) | 0.49 |
| 3000–<3500 | 227 (37.3) | 124 (36.9) | 103 (37.9) | | 23 (30.7) | 144 (37.7) | | 6 (40.0) | 161 (36.6) | |
| 3500–<4000 | 184 (30.3) | 98 (29.2) | 86 (31.6) | | 22 (29.3) | 118 (30.9) | | 2 (13.3) | 136 (30.9) | |
| ≥4000 | 78 (12.8) | 44 (13.1) | 34 (12.5) | | 8 (10.7) | 55 (14.4) | | 3 (20.0) | 60 (13.6) | |
| Missing | 61 | | | | | | | | | |
| Gestational age (wks) | | | | | | | | | | |
| Preterm (34–36) | 34 (5.6) | 22 (6.6) | 12 (4.4) | 0.63 | 8 (10.7) | 20 (5.2) | 0.12 | 2 (13.3) | 26 (5.9) | **0.01** |
| Early term (37–38) | 145 (23.9) | 82 (24.6) | 63 (23.1) | | 21 (28.0) | 80 (20.9) | | 8 (53.3) | 93 (21.1) | |
| Full term (39–40) | 339 (55.9) | 183 (54.8) | 156 (57.1) | | 37 (49.3) | 221 (57.9) | | 4 (26.7) | 253 (57.5) | |
| Late term (≤41) | 89 (14.7) | 47 (14.1) | 42 (15.4) | | 9 (12.0) | 61 (16.0) | | 1 (6.7) | 68 (15.5) | |
| Missing | 62 | | | | | | | | | |
| Birth mode and IAP | | | | | | | | | | |
| Vaginal, no IAP | 308 (50.7) | 169 (50.6) | 139 (50.9) | 1.00 | 35 (47.3) | 197 (51.4) | 0.18 | 8 (53.3) | 222 (50.5) | 0.96 |
| Vaginal IAP | 150 (24.7) | 82 (24.6) | 68 (24.9) | | 16 (21.6) | 98 (25.6) | | 3 (20.0) | 111 (25.2) | |
| Elective CS IAP | 68 (11.2) | 38 (11.4) | 30 (11.0) | | 7 (9.5) | 42 (11.0) | | 2 (13.3) | 47 (10.7) | |
| Emergency CS IAP | 81 (13.34) | 45 (13.3) | 36 (13.2) | | 9 (12.0) | 46 (12.0) | | 2 (13.3) | 60 (13.6) | |
| Missing | 62 | | | | | | | | | |
| Infant ethnicity | | | | | | | | | | |
| Asian | 77 (12.9) | 46 (13.9) | 31 (11.7) | 0.81 | 16 (21.3) | 41 (10.7) | **0.01** | 4 (26.7) | 53 (12.0) | 0.06 |
| First Nations | 47 (7.9) | 24 (7.3) | 23 (8.7) | | 9 (12.0) | 25 (6.5) | | 3 (20.0) | 31 (7.0) | |
| Caucasian | 439 (73.5) | 242 (73.1) | 197 (74.1) | | 46 (61.3) | 302 (78.7) | | 8 (53.3) | 339 (76.7) | |
| Other | 34 (5.7) | 19 (5.7) | 15 (5.6) | | 4 (5.3) | 16 (4.2) | | 0 (0.0) | 19 (4.3) | |
| Missing | 72 | | | | | | | | | |
| Season of birth | | | | | | | | | | |
| Summer (June–August) | 159 (26.0) | 93 (27.6) | 66 (24.0) | 0.31 | 11 (14.7) | 106 (27.5) | **0.02** | 3 (20.0) | 112 (25.3) | 0.64 |
| Other (October–May) | 453 (74.0) | 244 (72.4) | 209 (76.0) | | 64 (85.3) | 279 (72.5) | | 12 (80.0) | 331 (74.7) | |
| Missing | 57 | | | | | | | | | |
| Breastfeeding status at 3 mo | | | | | | | | | | |
| None | 93 (16.1) | 42 (13.3) | 51 (19.6) | **<0.01** | 14 (19.2) | 51 (13.4) | 0.3 | 4 (28.6) | 61 (13.9) | **0.05** |

**TABLE 1** (Continued)

| Participant characteristics | Prevalence Overall N (%) | Natural space <500m from residence | | | ≥1 atopic sensitization at 3 yrs | | | ≥2 inhalant atopic sensitizations at 3 yrs | | |
|---|---|---|---|---|---|---|---|---|---|---|
| | | Yes N (%) | No N (%) | P | Yes N (%) | No N (%) | P | Yes N (%) | No N (%) | P |
| Partial | 154 (26.7) | 74 (23.4) | 80 (30.8) | | 15 (20.6) | 103 (27.0) | | 6 (42.9) | 112 (25.6) | |
| Exclusive | 329 (57.12) | 200 (63.3) | 129 (49.6) | | 44 (60.3) | 227 (59.6) | | 4 (28.6) | 265 (60.5) | |
| Missing | 93 | | | | | | | | | |
| Household income | | | | 0.12 | | | 0.19 | | | 0.95 |
| <$50,000 | 73 (12.7) | 35 (10.9) | 38 (15.0) | | 8 (11.4) | 38 (10.3) | | 1 (6.7) | 45 (10.7) | |
| $50,000 to $99,000 | 192 (33.4) | 101 (31.5) | 91 (35.8) | | 20 (28.6) | 135 (36.6) | | 5 (33.3) | 149 (35.3) | |
| ≥$100,000 | 271 (47.1) | 165 (51.4) | 106 (41.7) | | 35 (50.0) | 180 (48.8) | | 8 (53.3) | 206 (48.8) | |
| Prefer not to say | 39 (6.8) | 20 (6.2) | 19 (7.5) | | 7 (10.0) | 16 (4.3) | | 1 (6.67) | 22 (5.2) | |
| Missing | 94 | | | | | | | | | |
| Maternal education | | | | **0.04** | | | 0.11 | | | 0.9 |
| Highschool or less | 54 (9.4) | 24 (7.5) | 30 (11.8) | | 1 (1.4) | 28 (7.6) | | 1 (6.7) | 27 (6.4) | |
| Some postsecondary | 210 (36.5) | 107 (33.3) | 103 (40.4) | | 23 (32.4) | 129 (35.0) | | 6 (40.0) | 146 (34.5) | |
| University degree | 240 (41.7) | 148 (46.1) | 92 (36.1) | | 40 (56.3) | 163 (44.2) | | 7 (46.7) | 195 (46.1) | |
| Postgraduate degree | 72 (12.5) | 42 (13.1) | 30 (11.8) | | 7 (9.9) | 49 (13.3) | | 1 (6.7) | 55 (13.0) | |
| Missing | 93 | | | | | | | | | |
| Maternal smoking | | | | **0.03** | | | 0.66 | | | 0.2 |
| Yes | 24 (4.2) | 8 (2.5) | 16 (6.2) | | 1 (1.4) | 8 (2.2) | | 1 (6.7) | 8 (1.9) | |
| No | 554 (95.9) | 312 (97.5) | 242 (93.8) | | 72 (98.6) | 363 (97.8) | | 14 (93.3) | 419 (98.1) | |
| Missing | 91 | | | | | | | | | |
| Maternal overweight/obesity | | | | 0.48 | | | 0.37 | | | **<0.01** |
| Yes | 225 (43.4) | 120 (42.0) | 105 (45.1) | | 26 (37.7) | 162 (43.6) | | 1 (7.1) | 186 (43.8) | |
| No | 294 (56.7) | 166 (58.0) | 128 (54.9) | | 43 (62.3) | 210 (56.5) | | 13 (92.9) | 239 (56.2) | |
| Missing | 150 | | | | | | | | | |
| Pets in home (pre or postnatal) | | | | 0.14 | | | 0.31 | | | 0.21 |
| Yes | 263 (54.7) | 139 (51.7) | 124 (58.5) | | 33 (50.0) | 186 (56.9) | | 5 (38.5) | 213 (56.2) | |
| No | 218 (45.3) | 130 (48.3) | 88 (41.5) | | 33 (50.0) | 141 (43.1) | | 8 (61.5) | 166 (43.8) | |
| Missing | 188 | | | | | | | | | |

aCS, cesarean delivery; IAP, intrapartum antibiotics. P-value from Chi-squared test. Bold indicates P < 0.05.

**TABLE 2** Crude associations between natural space exposure and atopic sensitization outcomes at ages 1 and 3 years[a]

| Atopic sensitization | | | Total N (%) | Natural space <500m from residence | | |
|---|---|---|---|---|---|---|
| | | | | Yes N (%) | No N (%) | P |
| Atopic sensitizations 1 yr | ≥1 | Yes | 91 (17.2) | 56 (19.0) | 35 (14.8) | 0.20 |
| | | No | 439 (82.8) | 238 (81.0) | 201 (85.2) | |
| | ≥2 | Yes | 28 (5.3) | 13 (4.4) | 15 (6.4) | 0.32 |
| | | No | 502 (94.7) | 281 (95.6) | 221 (94.6) | |
| Atopic sensitizations 3 yrs | ≥1 | Yes | 75 (16.3) | 50 (19.6) | 25 (12.2) | **0.03** |
| | | No | 385 (83.7) | 205 (80.4) | 180 (87.8) | |
| | ≥2 | Yes | 31 (6.8) | 14 (5.5) | 17 (8.3) | 0.23 |
| | | No | 427 (93.2) | 240 (94.5) | 187 (91.7) | |
| Food atopic sensitizations 1 yrs | ≥1 | Yes | 73 (13.8) | 45 (15.3) | 28 (11.9) | 0.25 |
| | | No | 457 (86.3) | 249 (84.7) | 208 (88.1) | |
| | ≥2 | Yes | 23 (4.3) | 11 (3.7) | 12 (5.1) | 0.45 |
| | | No | 507 (95.7) | 283 (96.3) | 224 (94.9) | |
| Food atopic sensitizations 3 yrs | ≥1 | Yes | 38 (8.3) | 24 (9.4) | 14 (6.8) | 0.32 |
| | | No | 422 (91.7) | 231 (90.6) | 191 (93.2) | |
| | ≥2 | Yes | 10 (2.2) | 5 (2.0) | 5 (2.4) | 0.73 |
| | | No | 450 (97.8) | 250 (98.0) | 200 (97.6) | |
| Inhalant atopic sensitizations 1 yrs | ≥1 | Yes | 23 (4.3) | 13 (4.4) | 10 (4.2) | 0.92 |
| | | No | 507 (95.7) | 281 (95.6) | 226 (95.8) | |
| | ≥2 | Yes | 1 (0.2) | 0 (0.0) | 1 (0.4) | 0.26 |
| | | No | 529 (99.8) | 294 (100.0) | 235 (99.6) | |
| Inhalant atopic sensitizations 3 yrs | ≥1 | Yes | 58 (12.6) | 36 (14.1) | 22 (10.7) | 0.28 |
| | | No | 402 (87.4) | 219 (85.9) | 183 (89.3) | |
| | ≥2 | Yes | 15 (3.3) | 4 (1.6) | 11 (5.4) | **0.02** |
| | | No | 443 (96.7) | 250 (98.4) | 193 (94.6) | |

[a]p-value from chi2 test. Bold indicates $P < 0.05$.

proximity to natural green spaces within 500m of their home residence (Table 1). For these infants, the average proportion of natural green space coverage was 7.1% (95% CI 6.2, 8.0). Natural green space-exposed infants were significantly more likely to be exclusively breastfed, more likely to have mothers with a university degree, and less likely to be born to mothers who smoked than infants who were not living in close proximity to natural green space (Table 1). At 3 years of age, 16.3% of infants were sensitized to at least 1 allergen and 3.3% were sensitized to ≥2 inhalant allergens (Table 1). Infants who had ≥1 allergen sensitization at 3 years were more likely to be male (61.3% vs 48.3%), of non-Caucasian ethnicity (61.3% vs 78.7%), and born outside of the summer months (38.6% vs 21.4%) than infants who were not sensitized at 3 years (Table 1). Infants who had ≥2 inhalant allergen sensitizations at 3 years were less likely to be born full term (26.7% vs 57.5%) and exclusively breastfed (28.6% vs 60.5%), and less likely to have mothers who were overweight or obese (7.1% vs 43.8%) than infants who had 1 or no inhalant allergen sensitization at 3 years (Table 1). Associations between other atopic sensitizations at 3 years and atopic sensitizations at age 1 year are presented in Tables S1 and 2.

We tested associations between natural green space proximity during infancy, and atopic sensitization to ≥1 or ≥2 allergens compared to no atopic sensitizations at 1 or 3 years of age. A greater percentage of infants exposed to a natural green space were sensitized to at least 1 food or inhaled allergen at 3 years; 19.6% compared to 12.2% of nonexposed infants ($P = 0.03$, Table 2). However, fewer infants who lived near a natural green space developed atopic sensitizations to ≥2 inhaled allergens ($P = 0.02$) (Table 2). Among infants living close to natural green spaces, 1.6% had ≥2 inhalant atopic sensitizations; this percentage was 5.4% in infants without natural green space proximity.

From logistic regression modeling, infants living within 500m of a natural green space were 72% less likely to have ≥2 inhalant atopic sensitizations at 3 years of age than infants without this exposure (OR = 0.28 [95% CI 0.09, 0.90]) (Fig. 1A). The crude

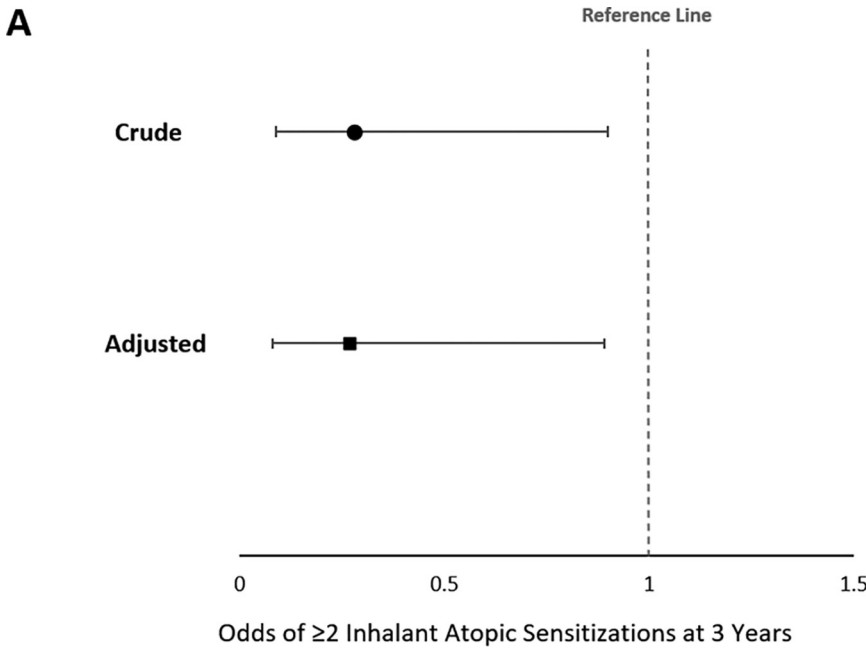

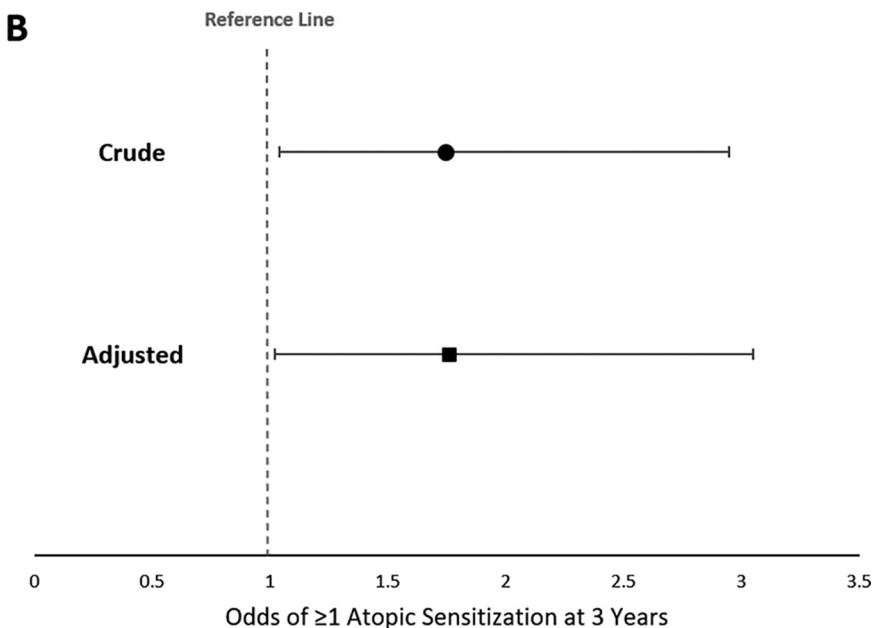

FIG 1 Forest plots portraying odds ratios and 95% CIs of developing atopic sensitization outcomes at age 3 years in infants exposed to natural space compared to those not exposed to natural space. (A) Odds ratios of having ≥2 inhalant atopic sensitizations at age 3 years in infants exposed to natural space compared to those not exposed to natural space <500m from home residence. Adjusted for pollution (tonnes NO2/km2) <3km from residence. (B) Odds ratios of having ≥1 atopic sensitization at age 3 years in infants exposed to natural space compared to those not exposed to natural space <500m from home residence. Adjusted for pollution (tonnes NO2/km2) <3km from residence.

odds ratio (OR) for developing ≥1 atopic sensitization at age 3 years following natural green space exposure during infancy was 1.75 [95% CI 1.04, 2.95]) (Fig. 1B). Following adjustment for $NO^2$ air pollution, these ORs did not change significantly, indicating independence of the natural green space-atopy association from air pollution in this study population (Fig. 1A and B). Adjusting models for other covariates (maternal education, ethnicity, household pets, and breastfeeding status) also did not significantly change these associations (Table S3). Natural green space exposure was not

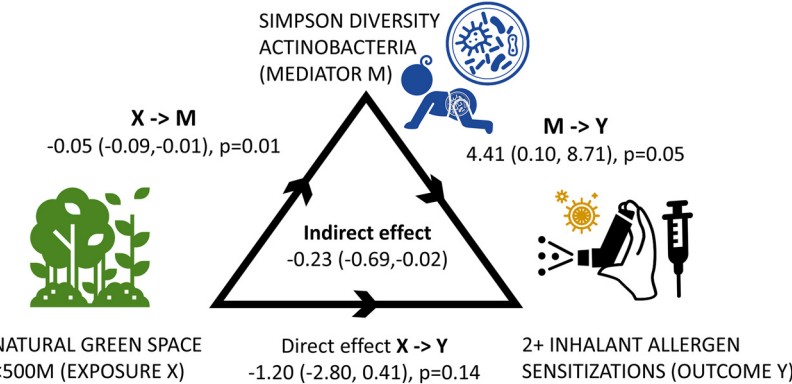

**FIG 2** Mediation of natural space <500m from residence and ≥2 inhalant atopic sensitizations at 3 years via Simpson diversity of Actinobacteria at 4 months. Beta estimates and 95% confidence intervals presented. 5,000 bootstrap. Acknowledgment of images: Eucalyp, Dima Lagunov, Parkjisun, Blair Adams.

significantly associated with any atopic sensitizations at 1 year, ≥2 atopic sensitizations at 3 years, sensitization to ≥1 or ≥2 food allergens at 3 years, or ≥1 inhalant allergen sensitizations at 3 years (Table S4).

Finally, utilizing 4-month fecal-sample microbiota profiles, we assessed whether total gut microbial and phylum-specific diversity (Chao1 species richness, Shannon and Simpson diversity metrics) mediated the observed statistically significant associations between natural green space exposure, and atopic or inhalant sensitization status at 3 years. None of the tested diversity metrics mediated these associations, with one exception. The relationship between living close to a natural green space and a substantially reduced likelihood of ≥2 inhalant atopic sensitizations at age 3 years was mediated by Simpson diversity of the phylum Actinobacteria (Fig. 2). In this mediation model, Actinobacteria Simpson diversity was (i) inversely related to natural green space proximity, with a beta-coefficient of -0.05 (95% CI: −0.09, −0.01), and (ii) positively related to multiple inhalant sensitization status, with a beta-coefficient of 4.41 (95% CI: 0.10, 8.71). The coefficient of the indirect effect of Actinobacteria Simpson diversity was 0.23 (95% CI −0.69, −0.02, Fig. 2), indicating that reduced Actinobacteria Simpson diversity mediated this protective effect of being exposed to natural green space.

## DISCUSSION

Among 458 infants, we found that proximity to natural green space within 500m of their home residence offered protection against developing sensitization to 2 or more inhalant allergens at age 3 years (OR = 0.28 [95% CI: 0.09, 0.9]). Further, Actinobacteria phyla diversity of gut microbiota at 4 months of age mediated this relationship. Similarly, Lehtimaki et al. found urban dwelling during infancy, when compared to rural residence, to raise future risk of aeroallergen sensitization; urban infants had a lower abundance of *Bifidobacterium* in their gut microbiota (24). Infant proximity to natural green space <500m raised the likelihood of sensitization in 3-year-old children to any allergen, but this association was not found to be mediated by the gut microbiome. Our findings are consistent with literature showing varying effects of natural green space exposure on atopy and asthma development in children by region, natural space exposure, and atopy/asthma outcomes. While previous studies have examined associations between any green space, including anthropogenic green space exposure, and childhood asthma in Canada (25), this study was the first to examine the impact of exposure to natural green spaces on atopic disease in a Canadian urban setting. Previously, we showed natural vegetation to be associated with greater species richness but reduced Simpson diversity of the Actinobacteria phyla in the young infant gut (23). Hence, we have further extended these findings by documenting that nature-associated changes to infant gut Actinobacteria have the capacity to mediate associations between

natural green space proximity and reduced multiple inhalant sensitizations at age 3 years. In the next paragraphs we further explain our findings, some of which may seem counter intuitive.

The gut microbiota of young, breastfed infants is dominated by infant-specific *Bifidobacterium* species, which results in higher Actinobacteria species richness but lower overall microbial diversity (26–29). At first glance, our results on the beneficial impact of reduced Actinobacteria diversity seem to contradict these observations or reported efficacy of bifidobacterial probiotics in preventing atopic disease (30, 31). However, lower values for the Actinobacteria Simpson diversity measure indicate an uneven distribution of species and enrichment with a few dominant bifidobacteria (32). Evidence is emerging that when a greater number of adult-associated species of *Bifidobacterium*, such as *B. catenulatum and B. adolescentis*, dominate gut microbiota at an early age, there is greater risk for atopy (31, 33–35). Similarly, we documented a positive correlation between greater Simpson diversity of Actinobacteria microbes and inhalant sensitization status in our mediation analysis. This correlation was reduced by proximity to natural green space and putatively, it was the outcome of enrichment with infant-predominant bifidobacterial species like *Bifidobacterium infantis* (28). Given the importance of early *Bifidobacterium* colonization and a low-diversity gut microbiome, the protective effect of natural vegetation on inhalant sensitization through enrichment with infant-specific *Bifidobacterium* warrants further investigation.

Contrary to the main findings is the observed positive association between natural green space proximity and sensitization to at least one allergen (food or inhalant) in 3-year-olds. A synthesis of the literature on the influence of green space finds inconsistency, with some studies showing protection from childhood atopy, while others reporting greater sensitization with green space exposure (18). No studies of food sensitization were identified in this systematic review. More recently, Peters et al. found a positive association between food allergy in infants and green space, even at medium levels of green space compared to low levels of green space (36). In our study, atopic sensitization status also included food sensitization and while not statistically significant, any food sensitization at age 3 years was also positively associated with natural green space exposure. Noteworthy is that infant gut microbiota did not mediate these positive associations, and tested confounding factors studied did not nullify them. Factors that we did not account for must be at play, such as household green space utilization patterns, microbial species level differences in vegetation or perhaps the contribution of fungi. Finally, having multiple atopic sensitizations, particularly to environmental aeroallergens, may be more predictive of true atopic asthma susceptibility (37–41).

This investigation had many strengths. The CHILD cohort study has an excellent retention rate and its longitudinal cohort study design is valuable in investigating the developmental origins of allergies and asthma. Since atopic disease was a major outcome of interest in the CHILD cohort study, comprehensive data were collected and clinical assessments conducted to identify atopic sensitization. The uPLVI map we employed has high resolution and provided a unique and more valid approach to classifying green space exposure by vegetation type and anthropogenic origin (forest versus parks) than satellite-mapped green space alone. This is relevant because varying types of outdoor spaces may harbor different flora and fauna, as well as support different microbes. We considered all natural vegetation (forest, grasslands) as natural green space exposure. Further, by looking at early-life atopic sensitization outcomes instead of atopic disease diagnoses later in life, the time course from exposure to outcome and the influence of any longitudinal covariates on our exposures and outcomes was minimized. This cohort's data collection is ongoing and will allow for these results to be compared to atopic disease outcomes at older ages in the future.

Despite the strengths, the present study also has some limitations. Firstly, the Edmonton site of the CHILD cohort is comprised of infants whose sociodemographics make them less likely to develop poor health outcomes, as infants were born primarily

healthy and at term from higher income families. This meant there was a low number of infants with atopic sensitization, making adjustment by confounding variables challenging due to insufficient statistical power. However, prevalence rates at ages 1 and 3 were comparable to the multicity CHILD Cohort Study population (41). Additionally, proximity to natural green spaces was used as a proxy for natural vegetation exposure. Infants living close to natural green space areas may not have spent much time outdoors, whereas those living further away may have been brought to these natural areas by their caregivers. Of the infants living <500m of a natural green space, the average proportion of natural vegetation coverage was 7.1%. It is unknown if a critical degree of vegetation coverage exists for the protective effect of nature proximity or if the protective effect rises with increasing proportions of natural vegetation coverage. Finally, while the association with natural green space was not diminished by air pollution, our air pollution measure was based on annual estimates of industrial air emissions from a national inventory (42). Despite these limitations, this study adds to an existing body of literature examining the effects of the natural environment on atopic disease development and offers unique perspectives on these effects in an urban Canadian setting.

Overall, our findings support the notion that natural green space proximity in infancy is important in reducing atopic sensitization to inhaled allergens and underpins possible mechanisms related to the gut microbiome. This is important from a public and personal health lens and, if confirmed, could have implications for green space protection and urban planning policies to reduce rates of asthma and atopy. Specifically, our findings could be used to support initiatives satisfying 5 out of 17 United Nations Sustainable Development Goals: Goal 3 (Good Health and Well-Being), Goal 11 (Sustainable Cities and Environments), Goal 13 (Climate Action), Goal 14 (Life Below Water), and Goal 15 (Life on Land) (43). Air pollution, climate change, and greenhouse gases worsen respiratory health, promote atopy, increase airborne pollen levels by changing weather patterns and events, and increase the severity and frequency of asthma (44–48). Considering this, understanding how green space can mitigate the impacts of ongoing climate change is of critical importance to reduce the already substantial individual and population-level burdens of asthma and other atopic diseases. Our findings further underscore the importance of environmental preservation and its beneficial role in protecting human health.

## MATERIALS AND METHODS

**Study population.** The study population included 699 infants who were enrolled in the CHILD (Canadian Healthy Infant Longitudinal Development) Cohort Study Edmonton site (www.childstudy.ca) and had complete uPLVI data (Fig. S1). They were singleton infants at ≥35 weeks of gestational age and a birth weight of ≥2500 g, born to pregnant women recruited in the second trimester. *In vitro* fertilized births were excluded, as were children born with congenital abnormalities or respiratory distress syndrome. Infants were enrolled at birth between 2009 and 2012 with consent provided by their parents. Of these 699 infants with uPLVI data, 530 and 460 infants had complete 1-year and 3-year atopic sensitization data, respectively, and were included in the atopy analyses. Of these, 287 subjects had complete 4-month gut microbiome, atopy, and uPLVI data and were included in the gut microbiome mediation analyses. Study approval was obtained from the University of Alberta Research Ethics Board; the CHILD Cohort Study was approved by the Hamilton Integrated Ethics Board (certificate number 07–2929).

**Exposure to natural environments.** The uPLVI map of the City of Edmonton was used to assess infants' exposure to natural spaces based on their postal codes provided at enrollment (23). Since the uPLVI map was unique to the city of Edmonton, this data linkage was only possible at the Edmonton site of the CHILD Cohort Study. In the present study, we defined natural space as being any natural area of green space origin. This definition of natural space included natural nonvegetated naturally occurring features, naturally wooded vegetated having ≥6% tree cover, naturally wooded vegetated having <6% tree cover, and wetland vegetated with minimum hygric moisture regimes. This definition excluded any land cover that was developed nonvegetated with anthropogenic origin or modified vegetated with anthropogenic origin. The average tree coverage found within Naturally Wooded Land Class (NAW) polygons was 78%; no NAW polygons had <40% tree coverage. As per a preceding study by Nielsen et al., in the present study a binary natural space exposure variable was created by summing the percent coverage of natural space <500m from home postal codes (23). Infants were considered exposed to a natural space if the percent of natural space coverage <500m from their postal code differed from 0% (23). Further methodological detail into this variable's generation and the uPLVI is provided by Nielsen et al.,

2020 (23). Fig. S2 shows the uPLVI map for the City of Edmonton. Fig. S3 shows an example uPLVI coverage by postal code used to generate nature exposure data and variables.

**Atopy assessment.** Epicutaneous SPTs for food and inhalant allergens were administered by CHILD study personnel as described in Tran et al. (41). Briefly, the SPT was considered positive if the allergen produced a wheal ≥2mm in diameter than the wheal elicited by the negative control, glycerin (41). Children with positive SPT(s) administered by the CHILD study or children whose caregivers provided SPT results from other physicians were considered to have atopic sensitization to the allergen(s) (41). Food allergens tested at 1 and 3 years were peanut, egg, soy, and milk. Inhalant allergens tested at 1 year were *Alternaria*, cat, dog, *Dermatophagoides pteronyssinus*, *Dermatophagoides farinae*, and cockroach. Inhalant allergens tested at 3 years were *Alternaria*, *Cladosporium*, *Penicilium*, *Aspergillus*, cat hair, dog hair, *Dermatophagoides pteronyssinus*, *Dermatophagoides farinae*, cockroach, tree mix, grass mix, weeds, and ragweed. Infant atopy was categorized by food allergens only, inhalant allergens only, or either. In addition, at each age, atopic children were additionally classified as being sensitized to one or more allergens or two or more allergens.

**Gut microbiota analysis.** Fecal samples were collected at a home visit at a mean infant age of 4.2 months (SD 1.2 months) and processed as described in Nielsen et al. and Tun et al. (23, 49). This time point for gut microbiota was used since it has been previously associated with atopic disease in the CHILD Study cohort (50). Briefly, fecal samples were collected at a home visit at 4-months-old and stored at −80° Celsius (23, 49). DNA was extracted, amplified, the V4 hypervariable region of the 16s rRNA gene was sequenced, and taxonomic classification was assigned based on 16s rRNA sequencing using RDP classifier constrained by the GREENGENES reference database (23, 49). The QIIME pipeline was used to summarize phylum, order, and family level data based on the relative abundances of bacterial operational taxonomic units (23, 49). Further methodological detail can be found in Nielsen et al. and Tun et al. (23, 49).

**Statistical analysis.** Stata 17.0 was used to conduct the analyses (StataCorp, 2021). Crude associations between nature exposure and atopy status, and between covariates and nature exposure or atopic outcomes were assessed using a $Chi^2$ test. Logistic regression was conducted to determine the association (odds ratio [OR], 95% confidence interval [CI]) between nature exposure (yes/no) and atopy status (yes/no) according to type of atopic sensitization, number of sensitizations (≥1 yes/no and ≥2 yes/no), and age at sample. A Directed Acyclic Graph (DAG) was created using DAGitty (http://dagitty.net/) based on a literature search to generate a minimal potential covariate adjustment set (Fig. S4). The minimal potential covariate adjustment set included presence of household pets (yes or no), maternal education (years), infant ethnicity (white or other), residential air pollution (geocoded as tons of $NO_2$ emitted/km$^2$ in area <3,000m from primary residence) (51), and breastfeeding status at 3 months of age (none, partial, or exclusive). To avoid over adjustment, only those covariates that changed the logistic regression OR estimate by >10% were included in the final adjustment set. Given these conditions, it was not necessary to adjust for any covariates in the final statistically significant models: "nature exposure and ≥1 atopic sensitizations at 1 year" model or the "nature exposure and ≥2 inhalant sensitizations at 3 years" model. The point estimates of the crude models were minimally changed even when all covariates in the minimal adjustment set were adjusted for (Table S3). Of final models yielding insignificant results, nature exposure and ≥2 atopic sensitizations at 1 year was adjusted for ethnicity, nature exposure and ≥2 food atopic sensitizations at 3 years was adjusted for air pollution, and nature exposure and ≥1 inhalant atopic sensitization at 1 year was adjusted for air pollution (Table S4).

**Mediation analysis methods.** The mediating effects of microbiota diversity measures at phylum level on the associations between exposure to nature and ≥1 atopic sensitization at 3 years and ≥2 inhalant sensitization at 3 years were tested using a STATA 17 command file created by Mike Crowson, Ph.D., The University of Oklahoma to testing a single mediator model involving a binary dependent variable in STATA (https://drive.google.com/file/d/1I2Zbi8Ux6HhVJQ6qmBhVnk9Mhxpsohlu/view). It has been designed to produce output consistent with Model 4 (involving a single mediator) from Andrew Hayes (see https://www.processmacro.org/index.html) Process macro. The criteria for X (natural green space) ->M (microbiota) and M (microbiota) ->Y (atopic sensitization) associations were required to be met before mediation analysis were conducted. Bootstrapping, a nonparametric resampling procedure (5000 bootstrap resamples) that improves reliability of estimates, was used to generate 95% CIs for coefficients in mediation models (52).

**Data access.** The STORMS (Strengthening the Organization and Reporting of Microbiome Studies, https://www.stormsmicrobiome.org/) checklist for reporting on human microbiome studies was used in the production of this manuscript. The STORMS checklist and other information can be accessed here: https://www.symbiotalab.com/downloads.

## SUPPLEMENTAL MATERIAL

Supplemental material is available online only.

**FIG S1**, TIF file, 0.3 MB.

**FIG S2**, TIF file, 1.5 MB.

**FIG S3**, TIF file, 0.9 MB.

**FIG S4**, TIF file, 0.1 MB.

**TABLE S1**, DOCX file, 0.04 MB.

**TABLE S2**, DOCX file, 0.04 MB.

**TABLE S3**, DOCX file, 0.02 MB.
**TABLE S4**, DOCX file, 0.02 MB.

## ACKNOWLEDGMENTS

The authors have no conflicts to declare. They would like to thank their funders for supporting this analysis, namely the Canadian Institutes of Health Research Microbiome Initiative (CIHR, 108028).

The CHILD Cohort Study was supported by both CIHR and the Allergy, Genes and Environment (AllerGen) Network of Centers of Excellence. We acknowledge that this work could not have been completed without the cooperation of all members, staff and participants of the CHILD Cohort Study. They include research staff, administrative staff, study families and participants, volunteers, lab technicians, statisticians, and clinical staff. The Canadian Institutes of Health Research Microbiome Initiative Emerging Team Grant (No. 108028) funded the infant gut microbial profiling.

We also thank the City of Edmonton for providing data on Urban Primary Land and Vegetation Inventory (uPLVI).

Mireia Gascon received support to conduct this work at the University of Alberta (Edmonton, Canada) from the Ministerio de Educación, Cultura y Deporte (Spanish Government) in the framework of the Programa Estatal de Promoción del Talento y su Empleabilidad en I+D + i, Subprograma Estatal de Movilidad, del Plan Estatal de I+D + I (José Castillejo grant). Mireia Gascon holds a Miguel Servet fellowship (Grant CP19/00183) funded by Acción Estratégica de Salud-Instituto de Salud Carlos III, cofunded by European Social Fund "Investing in your future."

Stuart Turvey holds a Tier 1 Canada Research Chair in Pediatric Precision Health.

We acknowledge support from the Spanish Ministry of Science and Innovation through the "Centro de Excelencia Severo Ochoa 2019- 2023" Program (CEX2018-000806-S), and support from the Generalitat de Catalunya through the CERCA Program.

Vienna Buchholz received support to conduct this work from the Alberta Innovates Summer Research Studentship and the Sears Undergraduate Summer Studentship.

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
