## [Reviewer comments · mSystems]

Natural Green Spaces, Sensitization to Allergens and the Role of Gut Microbiota during Infancy

Vienna Buchholz, Sarah Bridgman, Charlene Nielsen, Mireia Gascon, Hein Tun, Elinor Simons, Stuart Turvey, Padmaja Subbarao, Tim Takaro, Jeff Brook, James Scott, Piush Mandhane, and Anita Kozyrskyj

Corresponding Author(s): Anita Kozyrskyj, University of Alberta

Review Timeline:

Submission Date:	December 5, 2022
Editorial Decision:	December 13, 2022
Revision Received:	January 3, 2023
Accepted:	January 10, 2023

Editor: Suzanne Ishaq

Reviewer(s): The reviewers have opted to remain anonymous.

Transaction Report:

DOI: <https://doi.org/10.1128/msystems.01190-22>

December 13, 2022

Dr. Anita Kozyrskyj
University of Alberta
Edmonton
Canada

Re: mSystems01190-22 (Natural Green Spaces, Sensitization to Allergens and the Role of Gut Microbiota during Infancy)

Dear Dr. Anita Kozyrskyj:

Thank you for submitting your manuscript to mSystems. We have completed our review and I am pleased to inform you that, in principle, we expect to accept it for publication in mSystems. However, acceptance will not be final until you have adequately addressed the reviewer comments.

Preparing Revision Guidelines

Sincerely,

Suzanne Ishaq

Editor, mSystems

Journals Department
Reviewer comments:

Reviewer #1 (Comments for the Author):

I believe that the reviewer comments were addressed appropriately.

Reviewer #3 (Comments for the Author):

A study by Buchholz, V. et al. looked for the associations between natural green spaces in urban environments in the near-home environments of infants, their gut microbiota and later (early childhood) development of both food and inhalant sensitization with skin-prick test. A study has been already reviewed for the journal, and rejected after the first round, but editors decided to review article again due to contrasting first round reviews. I decided to only look the revised version and not to focus on earlier reviewer comments as I have not been involved in the process. My overall impression about this manuscript is that it approaches relevant research question with good reasoning: green spaces are beneficial for health, can urban green spaces influence health through early gut microbiota modulation? Needless to say, this topic is very demanding to study as many of the diseases originate from different reasons and also composition of gut microbiota is shaped by multiple variables. CHILD cohort is highly interesting, but not necessarily strong enough to study all these complexities. yet, this is observational study and should be considered as such. My major concern is that data has not been utilized very well (only diversity measures studied), especially if we consider that it has been submitted to microbiological journal. Much of the work has already been done in the earlier publication by the authors, but for reader it feels heavy to read also that one to get the full picture.

Specific comments:

I think the structure of the abstract could be improved. Now it heavily focuses on the background and methods are insufficiently described. Also, the results related to gut microbiome are limited considering the target journal.

For researchers like me, who are used to map-based, GIS-data, the description of the green space data is very superficial in the beginning of the manuscript. I don't think it is very scientific to describe data as "one-of-a-kind" without at least justifying such claim. Many cities are nowadays producing these types of datasets, even at the level of single tree. However, it is absolutely the strength of this manuscript that this urban green space data has been utilized in health research. I think it would be nice to see a map or other visual explanation of the environmental data showing why it is so great? Also, what is the difference of just using land cover data which also differentiates forest and parks? Moreover, the difference in canopy coverage was made on 6 %, which is very low canopy coverage considering that global land cover datasets estimates forest as areas with more than 30 % coverage.

Have you considered the possibility to look different types of green spaces? Of course, it might be that statistical power is too low as there was relatively little any green space in near-home environments of these children.

I found it a little hard to follow the findings on the lines 24 to 31. Was it so that you compared children with 1 and 2 or more sensitizations to each other? If so, why not to compare children without any to those with some? Does the green space proximity indicate yes/no or something more specific? You could consider presenting these findings in figure instead of table.

on row 32, you mention the 70 %. Where is this number coming from?

Why did you choose NO₂ from all the air pollution metrics to adjust for? Commonly PM_{2.5} is considered most important for the health.

Could you argue why adjustment did almost nothing for your findings in fig 1? Moreover, why these contrasting findings for 1 or more allergens are not mentioned in the abstract?

Could you argue why 4-month gut microbiota was considered important for the study?

Where mediation analyses adjusted?

Why only diversity indexes were studied?

You could consider the findings about food sensitization in relation to rural (greener) and urban (less green) in this manuscript (10.1016/j.jaci.2020.12.621), potentially helping your discussion.

I have one suggestion that authors could try: Neighborhood socio-economic status could be interesting and important confounding factor that has in previous research been associated to many aspects of health and wellbeing and greenness can be just a proxy of this. Adding that to your analysis focusing spatial aspects of health would improve manuscript a lot and would increase the fit of the story to the special issue run by editor.

mSystems01190-22

Buchholz et al. Natural Green Spaces, Sensitization to Allergens and the Role of Gut Microbiota during Infancy

Response to Reviewer #3:

I think the structure of the abstract could be improved. Now it heavily focuses on the background and methods are insufficiently described.

We have added a few sentences to the abstract on the methods used in the study.

Also, the results related to gut microbiome are limited considering the target journal. The mediating effect of infant gut microbiota in the association between nature proximity and atopic sensitization were tested and reported. Confirming gut microbiota pathways from an exposure to disease are very relevant to human health and align with the journal's mandate to 'achieve insights into the metabolic and regulatory systems at the scale of both the single cell and microbial communities.'

For researchers like me, who are used to map-based, GIS-data, the description of the green space data is very superficial in the beginning of the manuscript. I don't think it is very scientific to describe data as "one-of-a-kind" without at least justifying such claim. Many cities are nowadays producing these types of datasets, even at the level of single tree. However, it is absolutely the strength of this manuscript that this urban green space data has been utilized in health research. I think it would be nice to see a map or other visual explanation of the environmental data showing why it is so great?

Yes, of course. The Edmonton uPLVI natural vegetation and buffer zone maps were originally included in the supplementary information, ie. Figures S2 & S3.

Also, what is the difference of just using land cover data which also differentiates forest and parks? Moreover, the difference in canopy coverage was made on 6 %, which is very low canopy coverage considering that global land cover datasets estimates forest as areas with more than 30 % coverage.

The minimum of 6% canopy coverage followed the Government of Alberta's definition of naturally wooded vegetative areas. This minimum identifies 'open' forests, which are typical natural forests of northern Canada that include a mixture of wetlands, small trees and forest.

In reviewing the Edmonton's uPLVI map, there are no Naturally Wooded Land Class (NAW) polygons that have less than 40% coverage. The average tree coverage found in NAW polygons within the entire uPLVI database is 78%. We have now included these details in the Methods section (line 260).

Have you considered the possibility to look different types of green spaces? Of course, it might be that statistical power is too low as there was relatively little any green space in near-home environments of these children.

The Edmonton uPLVI map uniquely maps natural vegetation, which harbours different environmental microbes than man-made greenspace. This was our exposure of interest.

In our 2020 Environ Int paper (Nielson et al), we reported associations with infant gut microbiota with close proximity to natural spaces in Edmonton was unaffected by adjustment to man-made greenspace. We followed up on these findings in the current paper.

I found it a little hard to follow the findings on the lines 24 to 31. Was it so that you compared children with 1 and 2 or more sensitizations to each other? If so, why not to compare children without any to those with some?

We looked at associations between greenspace and 2 measures of atopy; ≥ 1 atopic sensitization vs none and separately, ≥ 2 atopic sensitizations vs none. We have revised these sentences to make this clearer in the results (lines 120-121) and also added additional words to explain this in the methods sections.

Does the green space proximity indicate yes/no or something more specific?

Natural green space proximity was yes/no variable as defined in our Methods (lines 263-67). It was the same variable used in our 2020 Environ Int paper.

You could consider presenting these findings in figure instead of table.

Indeed, the main results were presented in a forest-plot figure of Odds Ratios (Figure 1) and in a mediation triangle figure with colourful images (Figure 2). Only crude, descriptive findings were presented in a table.

on row 32, you mention the 70 %. Where is this number coming from?

We think you mean line 132. The 70% corresponds to the OR of 0.28, reported at the end of the same sentence. It is an alternate way of stating in lay words, the reduced odds of 0.28 (ie. $1 - 0.28$ or $\sim 72\%$ less odds). We have amended it to 72% to be clearer.

Why did you choose NO₂ from all the air pollution metrics to adjust for? Commonly PM_{2.5} is considered most important for the health. Could you argue why adjustment did almost nothing for your findings in fig 1?

NO₂ was an emissions-based area measure to which we had access for the city of Edmonton and which has been used in several published studies. We cited these papers and provided rationale in lines 294-298. Lack of difference in the Odds Ratio following statistical adjustment for NO₂ indicates independence of the association with natural green space or lack of precision of the NO₂ measure, the latter to which we alluded in the Discussion in lines 221-223.

Moreover, why these contrasting findings for 1 or more allergens are not mentioned in the abstract?

To meet the word count of the abstract, we omitted these findings. We have now added them to the abstract as follows: *We also found a positive association between nature proximity and sensitization to at least one food or inhaled allergen; this association which was not mediated by gut microbiota.*

Could you argue why 4-month gut microbiota was considered important for the study?

This was the first time point of fecal sample collection in the CHILD Cohort Study and has been associated with atopic sensitization in CHILD study publications (Azad et al. Clin Exp Allergy 2015).

Where mediation analyses adjusted?

No, they were not because none of the covariates affected the association between nature and atopy.

Why only diversity indexes were studied?

We followed up on statistically significant findings from our 2020 Env Int paper, where we found no associations with microbiota abundance phyla but did see differences according to microbial diversity within each phyla.

You could consider the findings about food sensitization in relation to rural (greener) and urban (less green) in this manuscript (10.1016/j.jaci.2020.12.621), potentially helping your discussion.

This is a relevant paper to cite. Thank you for bringing it to our attention. We have cited it in the first paragraph of the Discussion in lines 154-156 as follows: *Similarly, Lehtimäki et al found urban dwelling during infancy, when compared to rural residence, to raise risk of aeroallergen sensitization; urban infants had a lower abundance of Bifidobacterium in their gut microbiota.*

I have one suggestion that authors could try: Neighborhood socio-economic status could be interesting and important confounding factor that has in previous research been associated to many aspects of health and wellbeing and greenness can be just a proxy of this. Adding that to your analysis focusing spatial aspects of health would improve manuscript a lot and would increase the fit of the story to the special issue run by editor.

We agree this line of interrogation would nicely fit the story and individual-level SES measures (which are highly correlated with neighbourhood-level SES measures) were included as covariates in our study. Our individual-level measures of SES, ie. maternal income and education, were not associated with child atopy, nor did they change the Odds Ratios when added to models. Finally, our study infants were predominantly higher income, which we have noted as a limitation in lines 211-212.

January 10, 2023

Dr. Anita Kozyrskyj
University of Alberta
Edmonton
Canada

Re: mSystems01190-22R1 (Natural Green Spaces, Sensitization to Allergens and the Role of Gut Microbiota during Infancy)

Dear Dr. Anita Kozyrskyj:

Your manuscript has been accepted, and I am forwarding it to the ASM Journals Department for publication. For your reference, ASM Journals' address is given below. Before it can be scheduled for publication, your manuscript will be checked by the mSystems production staff to make sure that all elements meet the technical requirements for publication. They will contact you if anything needs to be revised before copyediting and production can begin. Otherwise, you will be notified when your proofs are ready to be viewed.

If you would like to submit a potential Featured Image, please email a file and a short legend to msystems@asmusa.org. Please note that we can only consider images that (i) the authors created or own and (ii) have not been previously published. By submitting, you agree that the image can be used under the same terms as the published article. File requirements: square dimensions (4" x 4"), 300 dpi resolution, RGB colorspace, TIF file format.

We recognize that the video files can become quite large, and so to avoid quality loss ASM suggests sending the video file via <https://www.wetransfer.com/>. When you have a final version of the video and the still ready to share, please send it to mSystems staff at msystems@asmusa.org.

Sincerely,

Suzanne Ishaq
Editor, mSystems

Journals Department
E-mail: mSystems@asmusa.org